# Analysis of the Classification Systems for Thoracolumbar Fractures in Adults and Their Evolution and Impact on Clinical Management

**DOI:** 10.3390/jcm11092498

**Published:** 2022-04-29

**Authors:** Bogdan Costachescu, Cezar Eugen Popescu, Bogdan Florin Iliescu

**Affiliations:** 1Department of Neurosurgery, “Gr. T. Popa” University of Medicine and Pharmacy, 700115 Iasi, Romania; costachescus@gmail.com; 2Department of Neurosurgery, “Prof. Dr. N. Oblu” Clinical Emergency Hospital, 700309 Iasi, Romania; ciepopescu@yahoo.com

**Keywords:** thoracolumbar, fractures, classification, management

## Abstract

Although they represent a significant chapter of traumatic pathology with a deep medical and social impact, thoracolumbar fractures have proven to be elusive in terms of a definitive classification. The ever-changing concept of the stability of a thoracolumbar injury (from Holdsworth’s two-column concept to Denis’ three-column theory), the meaningful integration of neurological deficit, and a reliable clinical usability have made reaching a universally accepted and reproductible classification almost impossible. The advent of sophisticated imaging techniques and an improved understanding of spine biomechanics led to the development of several classification systems. Each successive system has contributed significantly to the understanding of physiopathological mechanisms and better treatment management. Magerl et al. developed a comprehensive classification system based on progressive morphological damage determined by the following three fundamental forces: compression, distraction, and axial torque. Vaccaro et al. devised the thoracolumbar injury severity score based on the following three independent variables: the morphology of the injury, posterior ligamentous complex (PLC) integrity, and neurological status at the time of injury. However, there are limitations to the classification system, especially when magnetic resonance imaging yields an uncertain status of PLC. The authors review the various classification systems insisting on their practical relevance and caveats and illustrate the advantages and disadvantages of the most widely used systems with relevant cases from their practice.

## 1. Introduction

Spinal fractures represent a category of traumatic lesions with a significant impact both medical and social aspects of life. Within this category, thoracolumbar fractures (TLF) form a particular subcategory due to their location with specific biomechanics—the transition from thoracic physiologic kyphosis (fixed) to lumbar lordosis (mobile) [1]. Lesions at this level may have severe consequences such as complete/partial motor deficit, local pain or kyphosis [2].

The incidence of thoracolumbar traumatic lesions is variable dependent on the region and country and the number of cases ranges between 6.4 and 11.7/million/year in the United States [3] and 2.4/million/year in China [4], with a global incidence of 3/million/year, including osteoporotic fractures [5]. Within spinal fractures, thoracolumbar localization is present in more than half of the cases (58%) [6]. The most common localization is L1, followed by T12 and L2 [7]. Neurologic deficit (AIS A–AIS D) affects between 25–45% of the cases and 54% [6,7]. More importantly, the treatment of these fractures is still the subject of much controversy. There are three issues that have not yet found a definitive solution which are as follows: 1—a reliable classification/scale, 2—the best surgical approach [8].

In 2002, Mirza et al. defined the ideal criteria that have to be fulfilled in the classification of traumatic spinal lesions [9]. According to them, the terminology has to be clear, comprehensive, concise and descriptive, so as to clearly define each category; the lesion has to be described in terms of pathogeny and biomechanics (mechanism), to precisely indicate the severity and to guide the treatment; lesion characteristics have to defined both in terms of clinical and radiologic elements; the neurology has to be defined in terms of etiology, severity and manifestations; lesion morphology scaling has to take into account the severity of bony and ligamentous lesions, prognostic predictors, including natural history, treatment type and results, deformity risk and additional neurological deficits, to provide useful elements for future research.

Historically, multiple classifications and scales have been devised in order to optimally define the lesions as well as the treatment options. These systems evolved over time, considering not only the morphology of the lesion and mechanisms but also stability, neurology and PLC integrity.

In 2010, van Middendorp et.al. describe the ideal characteristics of a classification system for spinal traumatic lesions [10]. Ideal properties of spinal injury classification categories are, according to them, as follows: (1) clear definitions without ambiguity or freedom of interpretation; (2) all-inclusive and mutually exclusive; (3) clearly distinguishable representative graphic illustrations; (4) straightforward and practicable for daily use; (5) limited number of categories; (6) characterized by increasing grades of severity; (7) each (sub)category alphanumerically coded; (8) injury characteristics easily discernable on diagnostic images [10].

To summarize, based on the existing knowledge and the experience of the authors, a classification should comply to a sum of mandatory requirements. It should be simple and clear—constituent elements should be very well defined without category crossovers and should be easily usable in everyday practice. It also should be reproductible with minimal inter-and intra-observer differences. It must have a limited number of elements that are different and easily differentiable. It should include lesions localization, morphology, mechanisms and severity. It should offer a description of not only bony lesions, but also ligamentous involvement (Posterior ligament injury—PLC) allowing for vertebral stability evaluation as well as the prediction of segmental kyphosis. It also has to clearly define radiologically distinct lesions, and to provide a scale with an ascending score for gravity. It has to include the neurological deficit to provide guidance for the appropriate type of surgery and, ideally, for the timing of surgery, as well as indications for treatment, of either conservative or surgical approach types. Last but not least a comprehensive classification should include outcome prediction elements related to treatment, results, complications and it should allow a good standardization for trials and scientific studies.

## 2. Bohler Classification

The first classification of thoracolumbar traumatic lesions belongs to Bohler in 1930 [11]. In his classification he introduced lesion mechanism and morphology, describing 5 categories: compression fracture, flexion distraction injury (compression injury to the vertebral body and distraction injury to the posterior element), extension injury, shear fracture, and rotational injuries. He made no attempt to define spinal instability based on lesion morphology [12].

Sethi et.al., 2009, when looking at the existent classifications analyses, studied the Boehler classification and concluded that it is simple, easily usable, yet very descriptive, outdated, unvalidated, and cannot be predictive of outcome [13]. The same paper also evaluates other less used classifications, such as the Watson–Jones classification and the Nicoll, McAffee, Ferguson and Allen classifications [14,15,16,17]. Although these are considered simple and easy to use in practice, they are as a general rule not validated, they do not predict the outcome and as a result are not commonly used.

## 3. The Load Sharing Classification (LSC) of Spine Fractures

A classification based on a quantitative point system, The Load Sharing Classification (LSC) of Spine Fractures was published by McCormack and Gaines in 1994 [18]. This classification appeared as a result of the increasingly extensive use of pedicular screws, a technique described and used by Roy Camille, initially in spine fractures, then in other types of spinal pathologies [19,20,21,22,23]. The authors documented in certain cases the failure of short segment pedicular screws SSPS instrumentation systems—of one level above, or one level below—used for the treatment of three-columns fractures and fractures–dislocations in the thoracic and thoracolumbar areas, leading to kyphosis. This classification evaluates, based on a quantitative point system, the following elements: (1) the degree of kyphosis correction on lateral view, (2) degree of vertebral body comminution, (3) the apposition of the fracture fragment. Each element can be evaluated with 1 to 3 points, leading to an overall score of 3 to 9. As the score increases, the anterior support is more severely affected. SPSS is indicated for a score of 6 or lower. For a score greater or equal to 7, without dislocation (severe burst fractures) short-segment anterior corpectomy, strut-graft fusion and instrumentation is indicated. If dislocation is also present then, for the same score of 7 or greater, SPSS followed by an anterior approach (circumferential, 360 degrees) is indicated. Alternatively, LSPS (8 screws) can be used with the downside of supplemental mobility impairment. The authors underline from the beginning a couple of disadvantages of their classification such as a lack of evaluation of the mechanism of injury, ligament integrity, which in turn makes this score unusable for surgical decision making (the evaluation of the ligamentous system is essential in therapeutical decision). In conclusion, this classification is useful in making the choice between SSPS or LSPS, as well as the anterior approach (Figure 1).

Sethi et.al. advance the opinion that the Load Sharing Classification method of McCormack is simple, predicts the outcome of short instrumentation, but is not validated, and does not evaluate neurology and stability [13]. Radcliff et.al. conclude that the Load Sharing Score (LSS) cannot be used alone for surgical decision making for thoracolumbar fractures in the absence of a significant correlation between the neurological status and ligament integrity [14]. This caveat can lead to an incorrect choice of treatment, a low interobserver reliability for the proposed surgical treatment threshold score of 6, which may hinder communication and the continuity of care among providers, and surgical decision making at a score greater than 6 it is not reproducible, sensitive and specific.

Even more, both surgical techniques as well as instrumentation systems have evolved significantly in recent decades and, as a result, the LSS should be used carefully [15]. Stam et.al. in their systematic literature review published in 2020 conclude that, although an LSS equal or greater than 7 requires the reconstruction of anterior spinal support with or without SSPS, SSPS alone is sufficient and safe in these cases [16]. As a result, LSS loses its predictive value when faced with sagittal and posterior instrumentation failure.

In spite of all the above-mentioned disadvantages, LSS is valuable with regard to the issue of anterior vertebral support, taking into account the comminution and fragments apposition. These elements represent red flags for possible lesion progression towards chronic pain and segmental kyphosis, especially in cases treated conservatively.

## 4. Introduction of Columnar Concepts

A critical landmark in the evolution of thoracolumbar lesions classification was the introduction of the columnar concept. It was introduced in 1963 by Holdsworth as the “Two Column Concept” [17]. The author defined two parts for the spinal column, namely the anterior column consisting of the vertebral body, intervertebral disk and anterior and posterior ligaments, and the posterior column consisting of the pedicles, joint process, transverse apophysis, yellow ligament, spinous process and supra and interspinous ligaments. Column stability depends on the posterior column. This is the first instance that the role of PLC in column stability is noticed. Traumatic lesions are therefore split in stable—wedge compression fracture, burst fracture and unstable dislocations, i.e., extension fractures and dislocations rotational fracture-dislocation. The latter are produced through the following mechanisms: flexion, flexion-rotation, extension, extension-compression and translation. It is a classification that takes into account traumatic mechanisms and lesion morphology.

In 1983, Denis described the three columns system, which divides the Holdsworth anterior column in two, resulting a middle column which consists of the posterior half of the vertebral body, intervertebral disk and PCL (Figure 2). The middle column allows for a good differentiation between compression fractures that affect only the anterior column and burst fractures that involve both anterior and middle columns. Denis classifies thoracolumbar lesions as minor lesions—lesions of the transverse apophysis, spinous process, and isthmus (pars interarticularis) and major lesions—compression fracture, burst fracture, flexion-distraction injury (seat-belt injury), and fracture-dislocation [24]. Lesions are considered unstable if at least two columns are involved (e.g., a compression fracture with comminution is stable; a burst fracture with comminution produced by axial loading and compression of the vertebral body is unstable; the seat belt fracture that is the result of flexion and involves all three columns is unstable; a luxating fracture that involves all three columns and is a result of complex mechanisms of compression and rotation, is unstable). This classification has a series of disadvantages [25]. It does not evaluate the status of PCL that can lead over time at progressive instability, classifies all comminutive fractures as unstable (two columns involved), overlooks the neurology and provides no treatment guidelines.

## 5. Magerl Classification

Magerl et.al. described, in 1994, a new comprehensive classification for thoracolumbar fractures, the AO (Arbeitsgemeinschaft für Osteosynthesefragen) Classification or Magerl Classification, drawing on a study of 1445 fractures over a period of 10 years [26]. This is again a bi-columnar classification and is based on the radiographic morphology of the lesions (Rx and CT) as well as mechanism (compression, distraction, and axial torque). This classification uses a numeric system (3-3-3) and distinguishes three types of lesions referred to as A, B, and C. These are further divided, based on morphological criteria, into three groups which, in turn, have each a series of three sub-groups, adding supplementary specifications. Lesion severity and subsequently, the instability grow from A to C and from 1 to 3. Type A are compression fractures of the vertebral body without PLL involvement, which are stable or partially unstable. Type B are anterior and posterior element injuries with distraction. Comprising, as a rule, disk and ligamentous lesions, they have a surgical indication due to their poor healing potential. Type C anterior and posterior element injuries with rotation are unstable and require surgical treatment.

Sethi et al. consider the Magerl Classification method to be comprehensive. It is based on morphologic aspects, defining severity and instability, but is too complex, has moderate reliability and does not take into account the neurological deficit [13]. Furthermore, although it discusses instability, its definition is not clear, and almost all lesions have a degree of instability. Indeed, as a classification with 53 subtypes that is extremely detailed, with elements that are at times difficult to differentiate, makes its use in daily clinical practice quite difficult.

The reproducibility and reliability of Magerl Classification are not very high. If the lesions type and subtypes are considered, the interobserver agreement is fair (k = 0.32), also the intraobserver is also fair (k = 0.38) [27].

## 6. Thoracolumbar Injury Classification and Severity Score—TLICS

In 2005, Vaccaro et.al. proposed a new classification of thoracolumbar lesions with the Thoracolumbar Injury Classification and Severity Score (TLICS), trying to solve the disadvantages of previous classifications [28]. This classification can be considered a breakthrough, as it is the first one to take into account the neurological deficits. The other two independent predictors are lesion morphology and the integrity of the PLC. These elements are evaluated with a clinical exam, CT for morphology and MRI for ligamentous integrity (see Table 1).

Lesion instability is defined as a result that takes into account all three variables—morphology defines immediate mechanical stability (morphology of injury), the integrity of PLC—long term stability, and neurological status defines clinical stability.

A score of 3 or less suggests a conservative treatment, while a score of 5 or more suggests surgical treatment. Injury score 4 might be handled conservatively or surgically. Surgeon preference has to be taken into account. Additionally considered is patient preference after they are informed on the advantages, disadvantages and the risks associated with each treatment option.

A number of qualifiers have been described and they can influence the type of treatment independent of the TLICS score. These clinical qualifiers can change a surgical indication into a conservative one and contrariwise. Examples of qualifiers are local kyphosis, vertebral collapse, open fractures, overlying burns, multiple adjacent rib fractures, or an inability to brace. Additionally, general factors related to the traumatic event are also considered, such as sternum fracture, severe head injury, polytrauma or comorbidities—obesity, ankylosing spondylitis, rheumatoid arthritis, ankylosing spondylitis, osteoporosis and patient age.

Besides indicating the type of treatment, TLICS suggests as well the type of surgical approach, based on those three variables. This is another advantage of this score. As such, PLC injury requires, in general, a posterior approach, incomplete neurological deficit with neural compression from an anterior element requires an anterior approach, and the two elements combined require both posterior and anterior approaches [29].

Koh et.al., in 2010 consider that the intraobserver reliability of TLICS has a substantial effect on total score (k = 0.72) and injury morphology (k = 0.75), and almost perfect effect on neurological status (k = 0.96) and the integrity of PLC (k = 0.81) [30]. The interobserver reliability was substantially in agreement with injury morphology (k = 0.60) and the integrity of the PLC (k = 0.64), moderate agreement on total score (k = 0.57), almost perfect agreement on neurologic status (k = 0.91). The validity of the treatment chosen according to TLICS was 95%, sensitivity 98%, specificity 90%.

The same TLICS characteristics are less powerful according to Kaul [29]. Interobserver reliability was moderate for morphology (k = 0.43) and PLC integrity (k = 0.47), fair for total score (k = 0.29). Intraobserver reliability was moderate for morphology (k = 0.59) and PLC integrity (k = 0.55), and moderate (k = 0.44) for total score. Near perfect inter (k = 0.85) and intraobserver (k = 0.90) agreement was found for neurological status.

To summarize the literature data, the main advantages of TLICS are as follows comprehensive, defines stability considering all three factors (neurological status and PLC morphology being essential), gives objective criteria for treatment choice, establishes approach types, has acceptable reliability according to some of the authors, and has outcome predictive power [13,25,31]. Among the disadvantages, PLC evaluation is still difficult as it requires MRI (not available everywhere, sometimes difficult to perform in the context of polytrauma) and is the least reliable of the three variables of the score. When PLC lesion is not clear and is considered as “suspected/undetermined” at a score of 2, the treatment decision can shift towards surgical treatment even for lesions considered stable. A score of 4, when a lesion can be treated surgically or conservatively, is another disadvantage. Surgeon or patient preferences are not necessarily objective criteria. In some cases, even scores below three can pose decision problems.

## 7. AOSpine Thoracolumbar Spine Injury Classification System—ATLICS

In 2013, the AO Spine Classification Group (AOSCG) proposed a new classification, starting from Magerl Classification, without its disadvantages, simpler, more reliable and easier to use in clinical practice (Table 2) [31]. In this new classification, the three types of Magerl Classification are also reformulated into three groups, ordered ascendingly by severity as follows:

Type-A injuries: failure under axial compression of the anterior elements with intact posterior constraining elements with four subtypes; A1 and A2 are fractures of the superior plate or both plates, respectively, without comminution; and A3 and A4 are incomplete and complete burst fractures, respectively.

Type-B injuries: failure of the posterior elements (tension band or PLC injuries). Subtype B1 is the classic Chance fracture with a trajectory involving the pedicle and the vertebral body. B2 lesions comprise any lesion of the tension band, without bony involvement posteriorly. Lesions of the vertebral body are described separately (A subtypes).

Type-C injuries: failure of anterior and posterior elements with displacement. Subtype C1 affect the anterior spinal column through the disc or VB without displacement. However, the lesions can extend posteriorly, being characteristic in ankylosing spondylitis, and they are very unstable. Subtype C1—bony and disk and ligamentous lesions with displacement. Lesions of the vertebral body are described separately (A subtypes). Subtype C3 injuries result in a complete separation of the cranial and caudal parts of the spinal column.

This new classification, the AOSpine Thoracolumbar Spine Injury Classification System (ATLICS) was completed and modified in the same year, becoming a new instrument that combines Magerl classification and TLICS [32]. It includes three elements: injury morphology, neurological status and key modifiers. A number of changes concern lesion morphology. Type A0 is defined as comprising of fractures without clinical significance, at the level of transverse and spinous processes. Subtype B3 is the old type C1. The C subtype is represented by any spinal translation, without further subtypes.

Neurological status is described in the following fie levels: NO—neurologically intact; N1 transient neurological deficit, which is no longer present; N2 signs of radiculopathy; N3 incomplete spinal cord injury or cauda equina injury; N4 complete spinal cord injury.

Two modifiers are described to help in therapeutical decision making:

M1 is used in those cases where the status of PLC is unclear or cannot be determined. This is a useful modifier in those cases where a ligamentous lesion can decide in favor of surgery.

M2 is represented by patient comorbidities. As it was the case for TLICS clinical qualifiers, they can change the decision from surgical to conservative and contrariwise (ankylosing spondylitis, rheumatologic conditions, diffuse idiopathic skeletal hyperostosis, osteopenia/porosis, or burns affecting the skin).

This classification, as TLICS before it, brings a clearer neurological picture alongside lesion morphology and clinical modifiers. It is a simple classification into three types and nine subtypes with a clear-cut neurologic evaluation. Looking at ATLICS reliability, Kepler et al. reviewed the results of 100 spinal surgeons and concluded that overall interobserver reliability is moderate (k = 0.56), and overall intraobserver reliability is good (k = 0.68) [33].

Vaccaro et.al. provided, in 2016, a surgical algorithm based on ATLICS to guide therapeutical decision making—the thoracolumbar AOSpine injury score (TL AOSIS) [34]. Each lesion morphologic type, degree of neurologic deficit as well as the modifiers received a score.

**Table 2 jcm-11-02498-t002:** Original TL AOSIS classification as published by Vaccaro in 2016 [34].

Type A—Compression Injuries	Type B—Tension Band Injuries	Type C—Translational Injuries	Neurologic Status	Patient-Specific Modifiers
A0 0p A1 1p A2 2p A3 3p A4 5p	B1 5p B2 6p B3 7p	Translation 8p	N0 0p N1 1p N2 2p N3 4p N4 4p NX 3p	M1 1p M2 0p

## 8. Neurological Dysfunction

A key new aspect introduced by AOSpine TLICS (ATLICS) is the neurological status as a criterion in the evaluation of thoracolumbar fractures. Following the original TLICS classification experience, neurology was found to be essential in the clinical judgement of these cases. Any (new) neurologic dysfunction—regardless of biomechanical classification—shifts the attitude towards surgery.

## 9. Discussion

As a general rule, traumatic lesions with accompanying neurological deficit are considered surgical lesions. Controversy persists in those cases with normal neurologic status which could have, due to an insufficiently evaluated degree of instability, an unfavorable evolution, progressing over time towards local kyphosis, chronic pain and a potential new neurologic deficit. Of significant relevance is also the quality of life and return to normal daily activities. These lesions are represented in their majority by comminutive fractures types A3 and A4, with a TL AOSIS score of 1 to 3, as well as those in the grey zone of 4 and 5. The same is true for TLICS score of 1 to 3 and the grey zone of 4 and 5. This is one of the main reasons why additional evaluation criteria have to be defined on top of the above classifications. They should be used in controversial cases in order to reach a treatment decision. These influencers have the property of changing the decision bases in fracture considered a priori stable, so nonsurgical, to a surgical treatment. Not all comminutive fractures are alike, be they A3 or A5. They differ in terms of the degree of comminution independent of endplates involvement. If the comminution is important then it can evolve towards local kyphosis as a result of losing the anterior support.

As far as treatment is concerned, lesions with a score of 0–3 are treated conservatively, while those with 6 and higher treated surgically. A score of 4 or 5 (as TLICS 4) recommends either conservative or surgical treatment. The same disadvantage we saw in TLICS repeats here. Lesion with certain scores can be treated either way. In deciding the type of treatment, Vaccaro et.al. sent cases to a number of members of AO spine worldwide [33]. If 70% or more considered the lesion to be surgical than that was considered the optimal treatment. If less than 30% considered them surgical than those lesions should be treated conservatively. Sent cases included lesions that are highly controversial in terms of treatment, including A2N0 (TL AOSIS 2, as in case 1), different variants of A3 and A4, B1 and B2. The results showed conservative treatment for A2N0 and A3N0, and surgical treatment for A3 with transitory neurological deficit, radiculopathy, both with unclear PLC status (M1) A3N1M1, A3N2M1. A4 lesions with unclear PLC involvement and/or radiculopathy A4N0M1, A4N1M1, A4N2M0, A4N2M1 are also surgical. Type C lesions are clearly unstable and require surgery. We are still left with comminutive fractures that are not clearly specified which can pose treatment problems—A3 N0M1TL AOSIS 4, A4 N0M0 TLAOSIS 5, continuing the controversy of burst fractures without neurological deficit.

In 2014, Mattei et al. looked at those cases with a score of 3 or less, which have no surgical indication and have a good prognosis with conservative treatment alone [35]. For this specific group, he considers it to be of interest to complement the evaluation with the degree of comminution as described by LSC. Schnake meets these concerns by adding a series of modifiers that need to be accounted for in the management of thoracic and lumbar fractures that present an increased risk of local kyphosis such as severe comminution and loss of height of more than 50% [36]. Thoracic and thoracolumbar lesions have a greater risk of developing local kyphosis that lumbar fractures. Furthermore, a thoracolumbar or lumbar kyphosis of the same degree as a lumbar one is more significant as far as instability is concerned. To conclude, fractures with score equal or greater than 5 are definitely surgical lesions, while those with a score of 4 as well as those with a score equal or lesser than 3 have to be evaluated carefully using the above-mentioned modifiers for each specific situation.

In 2020, Park et al. retrospectively reviewed 328 patients with thoracolumbar spine lesions who all underwent MRI for optimal PLC assessment [37]. The goal was to evaluate the clinical validity of TLICS by comparing the management according to the TLICS and the applied treatment. In the conservative group (138patients), TLICS was 1–3 in 81.9%, 4 in 13%, more than 5 in 5.1%. The results show that the concordance between TLICS score and the therapeutical decision is relatively low. This is the result of general causes that contraindicated the procedure. For the cases with a score of 4 the main cause was the difficulty of decision making in comminutive fractures, as was the issue for those cases with a score lesser or equal to 3, even if the lesions seem stable.

Pantelaides et.al. published, in 2018, criteria for the selection of thoracolumbar surgical lesions—A1, A2 fractures when the compression is more than 50% and depending on the state of the posterior ligamentous complex; A3, A4 fractures depending on the fracture configuration; B and C fractures are unstable so surgery should be recommended [38]. These indications add A1 and A2 lesions, nonsurgical in theory, but there are no clear criteria for comminutive ones (A3 and A4). Mattei et.al. underlined the importance of the comminution extent and they suggested that it should be taken into account in the evaluation of certain comminutive fractures [35].

Local kyphosis as emphasized in LSC plays an important role in the evolution of the thoracolumbar traumatic lesions. Curfs et al. consider the following to be risk factors for post-traumatic kyphosis in incomplete burst fractures: type A3, the Cobb angle, Gardner angle and vertebral compression angle (over 20 degrees), T12, L1 localization and an age over 50 [39]. Shanke et.al. noted in their study, in 2008, that about 42% of all type B lesions were initially considered as type A, with no reference to the lesions of PLC [36]. They consider to be red flags local kyphosis over 15 degrees, important compression of vertebral cancellous bone with minimal reduced anterior vertebral height, and an additional element easily noticeable—anterior vertebral height reduced with more than 50%.

The Cobb angle is the most widely used measure for local kyphosis and is taken into account, alongside other modifiers in surgical decision making for those cases that, based on their TLICS or TL AOSIS score, should otherwise undergo a conservative treatment. Hitchon et al. analyzed a series of patients with a TLICS score of 2 that initially were treated conservatively. A quarter of these patients required surgery during the follow-up. In their view, increased local kyphosis, significant stenosis of the lumbar canal and a higher LSC would be the explanation for this phenomenon [40]. Tan et al., in their systematic review and metanalysis from 2021, looking at the rate and predictors of failure in the conservative management of stable thoracolumbar burst fractures, found that 9.2% of the conservatively treated patients have necessitated subsequent surgery [41]. Local kyphosis is considered an important predictor for the failure of the conservative treatment by some studies. However, it is still a very debated issue. Besides Cobb’s angle, the residual canal area, comminution and age could also play a role.

In 2021, WFNS Spine Committee considered as risk factors for post-traumatic kyphosis the following elements: more than 50 years of age, osteoporosis, disc injury above fractured vertebrae, three column fractures, T12-L1 level, PLC injury [42]. Age is a relevant factor that needs to be taken into account even before 50. At younger ages, rapid and complete recovery and return to an active life is essential. WFNS Spine Committee underlines the fact that surgical management leads to a faster return to work versus nonoperative management regarding returning to work [43].

## 10. Conclusions

Thoracolumbar traumatic lesions remain a controversial topic in modern orthopedics and neurosurgery in terms of classification and management. Although we do not yet have an ideal classification for a standardization of the treatment options, TLICS as well as AOSpine Thoracolumbar Spine Injury Classification System in tandem with TL AOSIS are simple, reliable, easy to use tools in daily clinical practice. For those cases that fall in the grey zone, associating influencers such as age, comminution level, local kyphosis and patient preference can lead to optimal treatment, be it surgical or conservative.

## Figures and Tables

**Figure 1 jcm-11-02498-f001:**
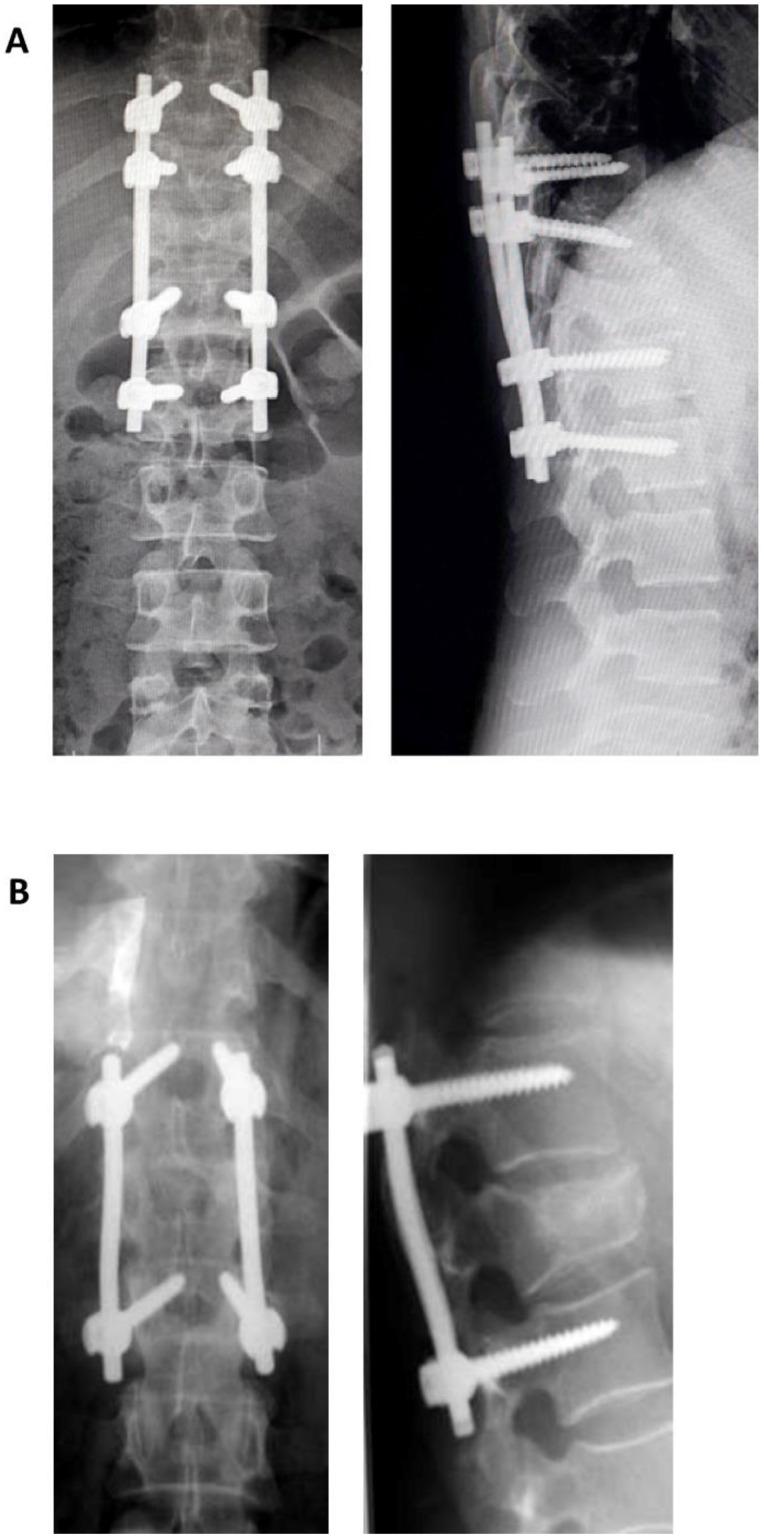
Postoperative X-ray showing a long segment pedicular screws (**A**) and short segment pedicular screws (**B**).

**Figure 2 jcm-11-02498-f002:**
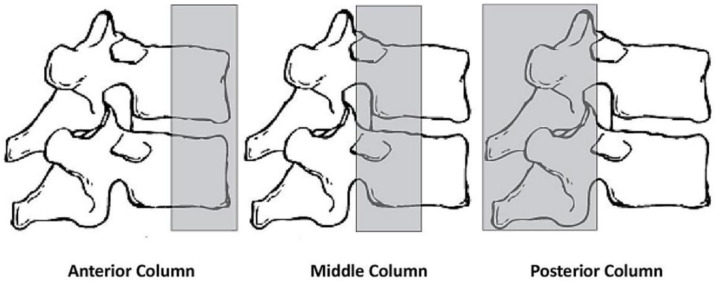
Schematic representation of Denis’ three column concept.

**Table 1 jcm-11-02498-t001:** Original TLICS score as published by Vaccaro et.al. in 2005 [28].

Morphology	PLC Integrity	Neurological Status
Compresion 1p Burst 2p Translation/Rotation 3p Distraction 4p	Intact 0p Suspected/indeterminate 2p Injured 3p	Intact 0p Nerve root injury 2p Complete cord injury 2p Incomplete cord injury 3p Cauda equina syndrome 3p

## Data Availability

Not applicable.

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
