# Peer review of "Analysis of the Classification Systems for Thoracolumbar Fractures in Adults and Their Evolution and Impact on Clinical Management"

_jcm, 2022, doi:10.3390/jcm11092498_

Round 1
Reviewer 1 Report
The topic of the manuscript is of neurosurgical and orthopaedic interest I, affecting daily spinal treatments. The paper is well outlined, even embodying the historical timelines and excellently researched. It can be also uselful used for teaching purposes.
- Introduction:
The paragraph is clearly outlines and provides an in-depth review to the subjects background. The question remains, whether the biomechanical impairments have not been covered widely enough by AO guidelines, whereas neurological deficitis should be proactively addressed through surgery. As for the timing of the treatment of neurological deficits, it is generally agreed upon, that decompression of neural structures should be performed within 72 hours of symptom onset. - Bohler Classification:
This section provides a concise review of the early classification and recognizes that there is no need to provide an in-depth analysis. - LSC of Spine Fractures:
The thorough argumentation for the introduction of load sharing classifications provides a superb read. - Columnar concepts:
The argumentation of the previous section is upheld for this paragraph, providing continuity for the understanding. Its contents should not be changed whatsoever. - Magerl Classification:
The section provides an in-depth and critical review for the AO classification, leading up to the refreshing information ahead in following sections. - TLICS:
This classification system, providing a more analytical approach to classify biomechanics and neurological deficits is imperative for the review of the subject, which the authors have correctly addressed. The included figures provide additional information and are appreciated, but could also be expanded further. - ATLICS:
The core classification for analysis and treatment of spinal fractures is excellently outlined within the paragraph and reviewed thoroughly. The case examples provide further information and comprehensiveness. - The discussion provides the reader with further insight and critical reviews, summarizing core concepts and rightfully concluding that more factors, such as age, comorbidities and type of surgical intervention should be improved for classification systems.
Author Response
Dear Reviewer,
Thank you for your thorough assessement of our review paper. We appreciate your comments and suggestions. As per your indications we added some new information and figures in order to make the paper clearer and more informative. Please note that we moved both cases as supplementary information.
Reviewer 2 Report
Dear Authors,
I like your review article and I think that it will shed light on the confusing classification systems of the thoracolumbar spine. However, to make the article more appealing to readers I think you could still improve it:
Abstract:
- Well written, no change needed
Introduction:
- Throughout the whole manuscript thoracolumbar / thoraco-lumbar / thoraco lumbar is written in 3 different ways. I suggest to write it together, as in the main title.
- Line: 43 timing of surgery is not discussed in the manuscript. It does not depend on the biomechanics of the fracture but primarily on sensomotoric disfunction. Please delete “3 – timing of surgery”
- Line 70: reproducable? – I recommend to restructure the sentence to “with minimal inter- and intraobserver differences”
Bohler Classification:
- Use past tense, when describing the introduction of this historic classification system
LSC + Magerl:
- Please provide the reader with a schematic picture. I recommend an image that depicts the column model by Denis. You have to keep in mind that a review article is often read by persons not completely familiar with the topic. Furthermore, it might serve as a thematic introduction for other research articles in the journal volume.
- Introduction of the columnar concepts
- again, we need some images to ease “the flow of reading” the article. Please, provide some schematic image that gives the reader a hint how to imagine the column concept.
- TLICS
- Line 198 (PLL = PCL))
- Due to the high number of abbreviations, provide a list of abbreviations in appendix A at the end of the manuscript.
- Introduce every introduction (LSPS?)
- Show a x-ray/CT as example for SSPS and LSPS
Case 1
- Delete both cases completely. What is the reason for adding these cases into a narrative review paper? If they were to be left in the article, it will be necessary to EXPLAIN ALL classification systems in regard to these cases. Case 1 would require a post-operative image/x-ray. But rather leave them out of the review, since they add nothing to the discussion about the “perfect classification system”. #
- AOSpine
- at the end of AOSpine neurology is discussed. This was never an issue in any other subheading/classification. Since the classifications are mainly biomechanical, I would introduce “Sensomotoric Dysfunction” as a separate subheading (e.g. 8). A short paragraph will be enough, because the review´s focus is on the biomechanics. The concluding information in this paragraph should be, that a (new) neurologic dysfunction – regardless of biomechanical classification – always requires fast surgery.
Case 2
- Same as Case 1, delete it. Maybe you can “re-use” the Figure 3 partially to explain the concept of posterior instrumentation, when referring to it in the main text of the manuscript. (SSPS etc)
Discussion:
- Line 377: “de” patients?
- 397: now we talk about Cobb angle for the first time in this paper. I recommend surgery, when compression is more than 15°. But this is under discussion in the literatur. There is plenty of literatur to that, none of the classifications addresses the development of a kyphoses after trauma. Please add more than just Ref 41 to this interesting issue. I suggest a small paragraph, about failure of conservative fracture management and why 15° or 20° kyphosis is a reason to perform surgery.
Author contribution:
In a narrative review is no methodology, no software (maybe pictures), no validation and no formal analysis – make sure that the contributions fits to the paper.
Author Response
Dear Reviewer,
Thank you very much for your through assessement of our review paper and especially for you comments and suggestions. We found them particularly useful in making the reviewr clearer and more informative. Please find the changes we made for each point bellow as italic comments:
Abstract:
- Well written, no change needed
Introduction:
- Throughout the whole manuscript thoracolumbar / thoraco-lumbar / thoraco lumbar is written in 3 different ways. I suggest to write it together, as in the main title.
- Line: 43 timing of surgery is not discussed in the manuscript. It does not depend on the biomechanics of the fracture but primarily on sensomotoric disfunction. Please delete “3 – timing of surgery”
- Line 70: reproducable? – I recommend to restructure the sentence to “with minimal inter- and intraobserver differences”
BFI, BC: Changed in text accordingly
Bohler Classification:
- Use past tense, when describing the introduction of this historic classification system
BFI, BC: Changed in text accordingly
LSC + Magerl:
- Please provide the reader with a schematic picture. I recommend an image that depicts the column model by Denis. You have to keep in mind that a review article is often read by persons not completely familiar with the topic. Furthermore, it might serve as a thematic introduction for other research articles in the journal volume.
BFI, BC: Introduced a schematic figure as Figure 2
- Introduction of the columnar concepts
- again, we need some images to ease “the flow of reading” the article. Please, provide some schematic image that gives the reader a hint how to imagine the column concept.
BFI, BC: Introduced a schematic figure as Figure 2
- TLICS
- Line 198 (PLL = PCL))
- Due to the high number of abbreviations, provide a list of abbreviations in appendix A at the end of the manuscript.
- Introduce every introduction (LSPS?)
- Show a x-ray/CT as example for SSPS and LSPS
BFI, BC: Made the due changes. Added an abbreviations list and introduced Figure 1 with postop x-ray of SSPS and LSPS
Case 1
- Delete both cases completely. What is the reason for adding these cases into a narrative review paper? If they were to be left in the article, it will be necessary to EXPLAIN ALL classification systems in regard to these cases. Case 1 would require a post-operative image/x-ray. But rather leave them out of the review, since they add nothing to the discussion about the “perfect classification system”. #
BFI, BC: Moved both cases as supplementary information
- AOSpine
- at the end of AOSpine neurology is discussed. This was never an issue in any other subheading/classification. Since the classifications are mainly biomechanical, I would introduce “Sensomotoric Dysfunction” as a separate subheading (e.g. 8). A short paragraph will be enough, because the review´s focus is on the biomechanics. The concluding information in this paragraph should be, that a (new) neurologic dysfunction – regardless of biomechanical classification – always requires fast surgery.
BFI, BC: introduced a new short subheader to emphasize neurology
Case 2
- Same as Case 1, delete it. Maybe you can “re-use” the Figure 3 partially to explain the concept of posterior instrumentation, when referring to it in the main text of the manuscript. (SSPS etc)
BFI, BC: Moved both cases as supplementary information
Discussion:
- Line 377: “de” patients?
BFI, BC: corrected
- 397: now we talk about Cobb angle for the first time in this paper. I recommend surgery, when compression is more than 15°. But this is under discussion in the literatur. There is plenty of literatur to that, none of the classifications addresses the development of a kyphoses after trauma. Please add more than just Ref 41 to this interesting issue. I suggest a small paragraph, about failure of conservative fracture management and why 15° or 20° kyphosis is a reason to perform surgery.
BFI, BC: Introduced a new paragraph, discussing the use of Cobb’s angle with some of the newest data.
Author contribution:
In a narrative review is no methodology, no software (maybe pictures), no validation and no formal analysis – make sure that the contributions fits to the paper.
BFI, BC: made the due changes
Round 2
Reviewer 2 Report
The authors did a great effort to improve the article and gave full response to all my objections. Therefore, I recommend to publish the revised version in JCM.